# The Association between Postgraduate Studies, Gender and Qualifying Dental School for Graduates Qualifying from UK Dental Schools between 2000 and 2009

**DOI:** 10.3390/dj5010011

**Published:** 2017-01-28

**Authors:** James Puryer, Joanna Selby, Joshua Layton, Jonathan Sandy, Anthony Ireland

**Affiliations:** School of Oral & Dental Sciences, Bristol Dental Hospital, Lower Maudlin Street, Bristol BS1 2LY, UK; js04171@gmail.com (J.Se.); joshualayton@mac.com (J.L.); Jonathan.Sandy@bristol.ac.uk (J.Sa.); Tony.Ireland@bristol.ac.uk (A.I.)

**Keywords:** UK, postgraduate, qualification, study

## Abstract

Various factors will influence a dental graduate’s decision to undertake postgraduate education and training, including encouragement from family, partners and staff at individual dental schools, although there is currently little information available regarding the number and distribution (by dental school) of recent dental graduates undertaking postgraduate studies. The aim of this study was to analyse data on postgraduate qualifications achieved by dentists who graduated from UK dental schools between 2000 and 2009 and relate this to graduate gender. Data were collected from the General Dental Council (GDC) in an anonymous electronic format, analysed and ordered by year of graduation, dental school, gender and type of postgraduate qualification. Nearly one-quarter (24%) of the dentists that graduated between 2000 and 2004 completed postgraduate studies, with more females (26%) than males (23%) obtaining further postgraduate qualifications. Overall, Bristol produced the largest proportion of graduates completing postgraduate study (39%) and of these the largest proportion of female graduates (45%). Glasgow produced the largest proportion of male graduates completing postgraduate study (37%). Membership of the Faculty of Dental Surgery (MFDS), one of the Royal Colleges, was the most popular postgraduate qualification obtained followed by Membership of the Faculty of General Dental Practitioners UK (MFGDP). This study provides insight into postgraduate studies undertaken by UK dental graduates. An increasing proportion of females are gaining Bachelor of Dental Surgery (BDS) qualifications and therefore the number of female dental graduates obtaining postgraduate qualifications is likely to increase further. This also suggests the male domination of the dental profession is likely to decrease.

## 1. Introduction

Each year, approximately 1300 dentists [1] graduate from 17 dental schools within the United Kingdom (UK). Following graduation, there are many avenues for career development and postgraduate studies available to them. Usually, this will involve returning to an academic institution or hospital based unit in order to gain further knowledge and skills within a specialist area of dentistry. This can lead to further registerable qualifications and eventual recognition by the General Dental Council (GDC) through the specialist list. The 13 dental specialties that are currently recognised by the GDC and the number of registrants per specialty [2] are shown in Table 1.

In addition to postgraduate qualifications leading to specialist status, there are a variety of further qualifications including Certificate, Diploma, Master’s and Doctorate degrees and the Royal Colleges’ Membership and Fellowship. These qualifications can be studied either part-time, full-time or in some cases by distance learning. In addition to furthering career development, the attainment of a postgraduate qualification can raise levels of professional satisfaction, especially amongst General Dental Practitioners (GDPs) [3]. Furthermore, studying for a postgraduate qualification helps to fulfil the Continuing Professional Development (CPD) requirements of the GDC, as well as ensuring that clinicians are up-to-date with knowledge and skills [4]. There are also research opportunities and a more structured approach now exists for those who wish to pursue academic careers including National Institute for Health Research (NIHR), Medical Research Council (MRC) and Wellcome Fellowships.

Various factors will influence a student’s decision to undertake postgraduate studies. A study carried out in the USA found that undergraduate experience plays a major role [5]. The strongest indicator as to whether a student/graduate is likely to specialise following graduation is the encouragement received from dental school staff, partners and family. It is likely that an individual student choosing to obtain postgraduate qualifications is highly influenced by the relationship with the specialist clinical staff they encounter during their undergraduate studies. A recent study of UK final-year dental undergraduates found that “having talent in a field” was the greatest factor influencing their wish to follow a specialist career [6]. Both of these studies suggest that school encouragement is an important influence on dental students’/graduates’ plans for postgraduate education and training.

The number of women applying for and pursuing a career in dentistry is continuing to rise. In 2014, 46% of UK dentists were female compared to only 27% in 1995 [1,7] and it is estimated that by 2020, over 50% of practising dentists will be female [8]. A study investigating differences between male and female dentists [9] found that 22% of women and 32% of men had obtained postgraduate qualifications. Although equal proportions of men and women had Masters’ degrees or dental diplomas, a higher proportion of males had Royal College Memberships, Fellowships or ‘other’ qualifications, and were registered specialists. This may partly be attributed to the fact that female dentists, in addition to working and studying, are often committed to bringing up children, with, on average 71% of female dentists reported as having children [10]. The needs of childcare are very influential on the potential future work patterns of female dentists [11]. In the UK, female dental students expect to take more time out of their careers to concentrate on childcare when compared to males [12]. Career breaks within dentistry will often make achieving postgraduate qualifications more difficult.

UK dental schools compete to attract the most able students and annual league tables are produced by various institutions including the Times Higher Education supplement, the Guardian, the QS World University Rankings and the Complete Universities Guide. The positions of each school appear to fluctuate from year to year [13] and the rankings are derived using various metrics including student satisfaction with courses, teaching and feedback, and career prospects after 6 months. As a result of these yearly variations, it can be hard to infer which schools are performing best over the longer term. As satisfaction at dental school plays a significant role in a dentist’s decision to undertake postgraduate studies [4], one possible method to examine this would be to study the proportion of students undertaking postgraduate study after gaining their first, primary dental degree. There is currently little information available regarding the number and distribution (by dental school) of recent dental graduates undertaking postgraduate studies.

## 2. Aims and Objectives

The aim of this study was to investigate postgraduate qualifications attained by dentists who graduated from UK dental schools between 2000 and 2009.

The objectives were:
To investigate the proportion of graduates from each UK dental school that undertake postgraduate education and training.To investigate whether there is a relationship between the types of postgraduate qualification achieved and the dental school where the dentist qualified.To investigate whether there is a relationship between gender and the likelihood of obtaining further qualifications.

## 3. Method

Information about dentists graduating from all UK dental schools between 2000 and 2009 was obtained from the GDC. This information was delivered in an anonymous format within the guidelines for data protection and included:
Number of graduates from each UK dental school per yearNumber of graduates from each UK dental school per year to go on to achieve a postgraduate qualificationPostgraduate qualification achievedSpecialty (if stated)Gender

This information was recorded on a spreadsheet and simple non-statistical analysis was carried out. Non-dental degrees such as Bachelor of Science (BSc) were not included in the analysis as these were most likely obtained prior to qualification and are thus not “postgraduate” study. The GDC ceased to collect detailed information on postgraduate qualifications in 2006, and so the majority of presented results relate to graduates from 2000 to 2004. In addition, the method of data collection by the GDC meant that it is not possible to identify an individual London school.

## 4. Results 

Almost four thousand (*n* = 3907) dentists qualified from all UK dental schools between 2000 and 2004, and an average of 24% (*n* = 965) accomplished some form of postgraduate qualification (Table 2). Bristol (39%) followed by Glasgow (35%) and then Dundee (34%) produced the highest proportion of graduates who obtained a postgraduate qualification.

The most popular qualification obtained was the Membership of the Faculty of Dental Surgery (MFDS), from one of the Royal Colleges which was awarded to 87% of all graduates who took a postgraduate qualification (Table 3). Overall, Dundee had the highest proportion of graduates choosing to study for MFDS, with 94% of their graduates obtaining this qualification, followed by Cardiff (93%) and Glasgow (93%). On average, 89% of female and 84% of male graduates who gained a postgraduate qualification attained MFDS. One hundred percent of Sheffield’s female graduates between 2000 and 2004 who undertook postgraduate study chose to gain the MFDS, followed by Dundee (95%) and Cardiff (94%). The latter also had the highest proportion of male students (93%) choosing to study for MFDS, followed by Glasgow (93%) and Dundee (92%). 

The second most popular postgraduate qualification obtained was the Membership of the Faculty of General Dental Practitioners UK (MFGDP (UK)), although the percentage of graduates choosing MFGDP (UK) as their postgraduate qualification was much lower (Table 4). Overall, Birmingham was the dental school with the highest proportion of graduates (27%) choosing to undertake the MFGDP (UK), comprising 26% females and 28% males who undertook postgraduate training. This was followed by Leeds (16%) and Belfast (14%). Belfast (21%) and Leeds (19%) also had a high percentage of females choosing to do MFGDP (UK), ranked second and third respectively. Sheffield had no female graduates who chose to gain the MFGDP (UK) qualification. Following Birmingham, Sheffield (16%) and London (12%) had the highest proportions of male graduates completing MFGDP (UK). Belfast had no male graduates from the five years in the study who attained MFGDP (UK).

Further qualifications gained by dentists who graduated between 2000 and 2004 are shown in Table 5. However, the amount of data is limited and thus it is not possible to identify any specific relationships between qualification, school or gender.

With the GDC discontinuing the collection of information about postgraduate qualifications post 2006, the information provided simply detailed the male and female graduate numbers from each dental school. Table 6 demonstrates the percentages of female and male graduates from each dental school between 2000 and 2009. On average, dental school graduates were 55% female, and 45% male. The highest proportion of female BDS graduates over this ten-year period was from Belfast (62%). By contrast, the dental school with the highest proportion of male graduates, and therefore lowest percentage of females, was Manchester with an average of 49% male graduates over this ten years period. Figure 1 shows the overall percentage of female graduates from UK dental schools over this ten-year period, with a linear trend that indicates a steady overall increase in female graduates between 2000 and 2009. 

Figure 2 shows the percentage of male and female graduates (2000 to 2004) from each dental school that obtained a postgraduate qualification. There is variation between schools in the percentage of female graduates obtaining a postgraduate qualification with Bristol having the highest (45%) and Belfast (15%) and Birmingham (15%) the lowest.

## 5. Discussion

This study provides objective data on postgraduate qualifications obtained by UK dentists who qualified between 2000 and 2009. There was a large range (12% to 39%) in the proportion of 2000 to 2004 graduates who undertook postgraduate qualifications from each of the 12 dental schools offering dentistry as an undergraduate degree during the period of study. Bristol had the highest overall proportion, and also the highest proportion of females completing postgraduate study. It appears that Bristol provides an undergraduate education that encourages further learning, possibly as a result of a high exposure to different specialties and the influence of specialist staff whilst an undergraduate. The reasons for this variation between schools are unclear and warrants further study, as it would be expected that undergraduates at all UK schools are exposed to different specialties. During the period 2004 to 2009, further dental schools opened within the UK, although graduates from these newer schools would not have been on the GDC register until 2010. Furthermore, it is not known if the dentists who undertook these postgraduate qualifications remain on the dental register as some may have retired, moved abroad or left dentistry altogether.

The fact that nearly one-quarter (24%) overall of all UK female graduates go on to undertake further study is noteworthy, considering that postgraduate education usually takes place during the childbearing years of a woman’s life. This trend for increasing numbers of female dentists to undertake formal postgraduate qualifications has been previously been reported overseas. A study conducted in Saudi Arabia [14] found that 54% of female respondents had completed formal postgraduate education. Over one-half (58%) of these females had a Master’s degree, and 45% of these were obtained abroad. Most women complete postgraduate training before starting a family, whereas men are more inclined to have children prior to or during postgraduate study [9]. Our own study supports the previous findings that newly qualified female dentists are undertaking postgraduate study early in their professional careers, possibly prioritising careers over family. The reasons for the variation between schools of the proportion of female graduates undertaking a postgraduate qualification are unknown and warrants further research. It is also evident that an increasing number of females are qualifying from all dental schools, and this trend appears to be rising, with 58% of additions to the 2014 UK dental register being female [5] compared to 53% between 2000 and 2014. This confirms that dentistry is becoming an increasingly popular career choice for women, and as a result, it is likely that there will be an increase in the number of female specialists within the UK dental workforce [15]. A Canadian study [16] found that the average dental working career length differs between males (35 years) and females (20 years), and moreover, women on average work fewer hours than males [17], and take more career breaks [18]. This supports an earlier New Zealand study [9] which also found that greater numbers of female than male dentists took career breaks, a large proportion being taken for childcare reasons. Career breaks have been associated with shorter working hours on return to the profession, and it is thought that female dentists who take a career break have a working life that is approximately 25% shorter than dentist who do not [19]. In addition, the percentage of female undergraduates who intend to work part-time 5 and 15 years after qualifying is higher than that of male undergraduates [12,20]. However, this may be offset by the fact that as greater number of female dentists undertake formal postgraduate qualifications, usually at significant personal financial cost, these dentists may be more likely to prolong their working careers in order to recoup some of this expenditure. This overall trend for an increase in the number of female dentists on the UK register may ultimately have an impact on access to dentistry. Relevant stakeholders including the General Dental Council and the Department of Health should be aware of this continuing trend such that the future UK dental workforce can be planned accordingly, along with ensuring that sufficient postgraduate courses are available to meet demand. When considering the types of postgraduate qualification attained, MFDS and MFGDP (UK) were the most popular, with 87% of postgraduates obtaining the MFDS. This is perhaps not surprising. Firstly, both of these qualifications can be obtained following graduation without the need for prior additional qualifications. Secondly, MFDS, and the more recent Membership of the Joint Dental Faculties (MJDF) offered by the English Royal College of Surgeons, are becoming important pre-requisites to undertake specialist training within the UK, and it would now be unusual to be appointed to a specialist training position without either of these qualifications. There is good evidence that current dental undergraduates have a greater desire to undertake specialist training than previously [11,12], and it could be expected that these two qualifications will become even more popular with new graduates. Recent research in the UK amongst final-year undergraduates found that “having a talent in a field” was the greatest reason for aspiring to take specialist training, followed by “reward” and “financial reasons” [6]. However, not all dentists who undertake postgraduate qualifications will undergo specialist training, and many of these dentists will continue to work in primary care. Although the data relating to additional qualifications (up to 2006) are limited, it is clear that the Membership in Orthodontics (M.Orth) is a popular postgraduate qualification. This is not surprising as there are more registered orthodontic specialists than in any other GDC recognised specialty, and this is likely to continue as orthodontics has been reported to be the most popular intended specialty of current UK undergraduates [12]. There is evidence that the UK needs a far greater number of specialists in order to meet the dental needs of the population [21,22] and the increasing desire for dentists to specialise is encouraging.

The present study provides an insight into the direction that dentists graduating between 2000 and 2004 have taken in terms of postgraduate study, both in terms of qualifications gained and also the distribution of these qualifications according to dental school. Unfortunately, as the GDC stopped collecting information on postgraduate qualifications in 2006 on the basis that they do not quality assure postgraduate programmes, the graduates in this study had, at most, six years to gain their postgraduate qualification. The data from 2000, therefore, shows a greater variety of postgraduate qualifications in comparison to 2004, where there was only a two-year window for a graduate to have completed postgraduate qualification. 

## 6. Conclusion

The aim of this study was to gain information on postgraduate qualifications attained by UK dentists who graduated between the years of 2000 and 2009. It is encouraging that nearly one-quarter of all dentists qualifying between 2000 and 2004 completed some form of formal postgraduate studies. The GDC data suggest an overall increasing proportion of females are studying dentistry as a career and this may ultimately have an effect on the UK dental workforce and access to dentistry. There is variation in the proportion of graduates from each of the dental schools in this study who undertake postgraduate qualifications, although the reasons for this are unclear and require further research.

## Figures and Tables

**Figure 1 dentistry-05-00011-f001:**
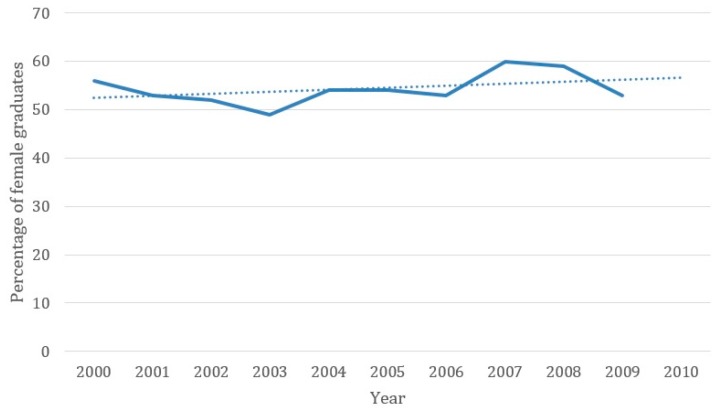
The percentage of female graduates from all dental schools 2000 to 2009 with linear trend line.

**Figure 2 dentistry-05-00011-f002:**
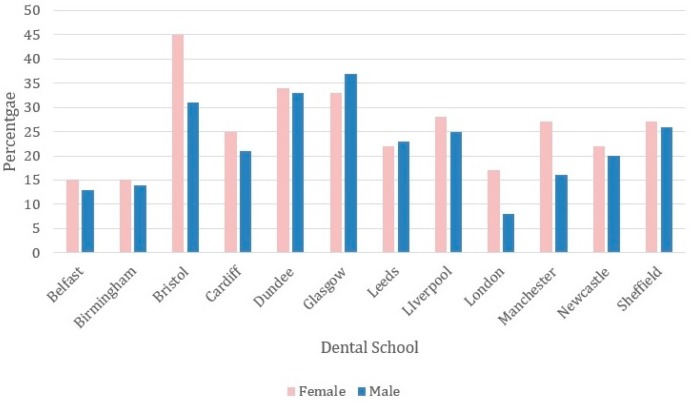
The percentage of male and female graduates (2000 to 2004) from each dental school who obtained a postgraduate qualification.

**Table 1 dentistry-05-00011-t001:** Dental specialties recognised by the General Dental Council (GDC) and number of registrants (March 2015).

Specialty	Number of Registrants
Dental & Maxillofacial Radiology	26
Dental Public Health	115
Endodontics	269
Oral & Maxillofacial Pathology	30
Oral Medicine	69
Oral Microbiology	8
Oral Surgery	738
Orthodontics	1342
Paediatric Dentistry	231
Periodontics	352
Prosthodontics	438
Restorative Dentistry	310
Special Care Dentistry	309

**Table 2 dentistry-05-00011-t002:** Percentage and ranking of graduates from each UK dental school between 2000 and 2004 who completed postgraduate study.

School	Total Graduates (*n*)	Overall	Rank	Female	Rank	Male	Rank
Belfast	152	14%	10	15%	11	13%	11
Birmingham	309	13%	11	15%	12	14%	10
Bristol	240	39%	1	45%	1	31%	3
Cardiff	276	22%	7	25%	7	21%	7
Dundee	236	34%	3	34%	2	33%	2
Glasgow	341	35%	2	33%	3	37%	1
Leeds	252	22%	6	22%	9	23%	6
Liverpool	242	12%	12	17%	10	8%	12
London	961	27%	5	28%	4	25%	5
Manchester	310	21%	9	27%	6	16%	9
Newcastle	335	21%	8	22%	8	20%	8
Sheffield	253	27%	4	27%	5	26%	4
Total	3907	24%		26%		23%	

**Table 3 dentistry-05-00011-t003:** Percentage and ranking of graduates from each UK dental school between 2000 and 2004 who gained Membership of the Faculty of Dental Surgery (MFDS) out of all graduates who completed postgraduate study.

School	Total Number Completing Postgraduate Study (*n*)	Overall	Rank	Female	Rank	Male	Rank
Belfast	21	81%	9	79%	10	86%	5
Birmingham	41	73%	12	74%	12	72%	12
Bristol	90	85%	7	90%	7	76%	11
Cardiff	61	93%	2	94%	3	93%	1
Dundee	79	94%	1	95%	2	92%	3
Glasgow	96	93%	3	94%	4	93%	2
Leeds	56	79%	11	74%	11	84%	6
Liverpool	30	80%	10	80%	9	80%	9
London	220	84%	8	87%	8	81%	8
Manchester	66	86%	6	92%	6	79%	10
Newcastle	72	90%	5	93%	5	88%	4
Sheffield	69	91%	4	100%	1	81%	7
Total	965	87%		89%		84%	

**Table 4 dentistry-05-00011-t004:** Percentage and ranking of graduates from each UK dental school between 2000 and 2004 who gained Membership of the Faculty of General Dental Practitioners (MFGDP) (UK) out of all graduates who completed postgraduate study.

School	Total Number Completing Postgraduate Study (*n*)	Overall	Rank	Female	Rank	Male	Rank
Belfast	21	14%	3	21%	2	0%	12
Birmingham	41	27%	1	26%	1	28%	1
Bristol	90	5%	8	5%	6	6%	8
Cardiff	61	3%	11	3%	8	4%	10
Dundee	79	5%	9	5%	7	6%	9
Glasgow	96	4%	10	2%	11	7%	7
Leeds	56	16%	2	19%	3	12%	4
Liverpool	30	10%	4	10%	4	10%	6
London	220	9%	5	6%	5	12%	3
Manchester	66	6%	7	3%	9	10%	5
Newcastle	72	3%	12	3%	10	3%	11
Sheffield	69	7%	6	0%	12	16%	2
Total	965	87%		89%		84%	

**Table 5 dentistry-05-00011-t005:** Total number of additional qualifications awarded to dentists graduating between 2000 and 2004.

Qualification	Number
Diploma in Dental Public Health (DDPH)	1
Diploma in Postgraduate Dental Studies (DPDS)	6
Fellowship in Dental Surgery (FDS)	4
Bachelor of Medicine, Bachelor of Surgery (MB BS or MB ChB)	3
Master of Clinical Dentistry (MClinDent)	1
Master of Clinical Dentistry—Prosthetics (MClinDent)	3
Membership in Orthodontics (M Orth)	9
Diploma of Member of the Faculty of Dentistry (MFD)	2
Membership in Paediatric Dentistry (MPaed Dent)	2
Membership in Restorative Dentistry (MRD)	2
Master of Science (MSc)	2
Master of Science—Conservative Dentistry (MSc Con Dent)	1
Master of Science—Endodontics (MSc End)	2
Master of Science—Implant Dentistry (MSc ImpDent)	2
Master of Science—Orthodontics (MSc Orth)	4
Master of Science—Prosthetic Dentistry (MSc ProsthDent)	1
Master of Science—Sedation & Special Care Dentistry (MSc SedSpCaDent)	1
Membership in Special Needs Dentistry (MSND)	1
Doctor of Philosophy (PhD)	1

**Table 6 dentistry-05-00011-t006:** Percentage and ranking of female and male graduates from each UK dental school between 2000 and 2009.

School	Female	Rank	Male	Rank
Belfast	62%	1	38%	12
Birmingham	56%	3	44%	10
Bristol	54%	8	46%	5
Cardiff	52%	10	48%	3
Dundee	56%	6	44%	7
Glasgow	56%	4	44%	9
Leeds	56%	5	44%	8
Liverpool	53%	9	47%	4
London	55%	7	45%	6
Manchester	51%	12	49%	1
Newcastle	52%	11	48%	2
Sheffield	57%	2	43%	11
Total	55%		45%

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
