# Peer review of "The Association between Postgraduate Studies, Gender and Qualifying Dental School for Graduates Qualifying from UK Dental Schools between 2000 and 2009"

_dentistry, 2017, doi:10.3390/dj5010011_

Round 1

Reviewer 1 Report

This paper concerns the issue of gender differences in postgraduate educations within UK. This is an important question as the number of male and female dentists with postgraduate qualifications can have consequences for the dental services and for the dental profession. The paper is well written and with a clear structure.

Unfortunately I find the contribution of gender aspect of dentistry unclear and potential consequences of an increased proportion of female dentists with postgraduate qualifications is not discussed in a sufficient way. More previous studies that concern effects of changing gender proportions in dentistry should be used in the discussion of the results of this study. It should be interesting to know how the results could be used in discussions of implications for undergraduate and postgraduate education and workforce planning.

Furthermore, the purpose of the article is not clear as there are other objectives than those that are related to the issue of gender. The relevance of these objectives in relation to the gender aspect have to be clearer if gender should be the main focus, which the title of the article suggest, or the objectives should be revised. Additionally, the abstract has to be rewritten if the gender should be the focus of the article. Otherwise the title has to be changed.

Author Response

Thank you very much for your very favorable  review of our paper.

We have made some changes as per your recommendations. Specifically we have:

a) added to the Discussion to expand on the implications of increasing numbers of female postgraduates

b) changed the title of the paper to reflect the additional paper objectives

I hope that you will view these changes in a positive light and give further support for the publication of this paper.

Kind regards

Reviewer 2 Report

I have with great interest read this paper about the proportion of graduates from dental schools that undertake postgraduate education and training. I have some observations that should receive the attention of the authors.

1. The level is appropriate to our readership.

2. Page 3, line 90: The objectives of this study include five research questions. I would have wished to see pone to three main research questions. Although it is valuable for studies to evaluate several aspects, it is important to identify a small set of primary research questions in advance. In addition, please list the objectives in the same order they were reported in the results section. 

3. Page 3, row 121: What is your sample size? How many dentists qualified from all UK dental schools between 2000 and 2004? How many accomplished some form of postgraduate qualifications? Please report also the frequencies, not only percentages.

4. Page 3, row 114: The statistical intensity of this paper is not high. Which summary measures were used? Help your readers and clearly document all data analysis methods.  .

5. Results section: The authors have reported their findings by including six tables and two figures. Figure 2 shows a time series. Please, use a line plot, not a bar plot.    

6. Discussion section: Although this study gained information on postgraduate qualifications from one country, I feel that many of the points raised are applicable to other settings in many countries. For publication in an international journal, it would help if the authors compare their findings with findings from other countries with different regulations and culture regarding university level postgraduate studies. 

Author Response

Thank you very much for your very favorable  review of our paper.

We have had opportunity to address all of your suggestions for improvement and we have specifically:

a) Reduced the number of Objectives from 5 to 3

b) Re-ordered the Objectives appropriately

c) Added the sample size and numbers of postgraduates

d) Clarified that statistical analysis was basic

e) Changed Figure 2 from a bar plot to a line plot

f) Added two further references relating out findings to studies carried out in other countries

I hope that you will view these changes in a positive light and give further support for the publication of this paper.

Kind regards

Reviewer 3 Report

Dear Authors, 

Thank you for the oppotunity to read your interesting paper. I find the paper relevant with an interesting perspective.

However I do have some comments.

What is the total number of dentists that qualitied in the period 2000 - 2009 and 2000 - 2004? 

Table 2, 3 and 4 are not selvexplanatory. It would be of great help to the reader with and overall number of qualifying dentists and the number dentists qualifying from each university.

add total number of participants included in the tables and figures, as well as the number of participants from each university

Figure 1 show that same results as the left percentage collumn in table 2. it would be better to precent either the table or the figure.

in figure 2 you argue that there is a trend for increase in female graduates, did you test for this trend?

did you consider testing for gender difference in qualifying dentists? 

Author Response

Thank you very much for your very favorable  review of our paper.

We have had opportunity to address all of your suggestions for improvement and we have specifically:

a) Added the total numbers of qualified dentists

b) Added numbers to Tables 2, 3 and 4

c) Removed Figure 1 (as this was duplicating Table 2) and renumbered Figures 2 and 3 accordingly

The trend for increasing numbers of female dentists was obtained from the automated chart 'trend line' rather than direct statistical analysis

I hope that you will view these changes in a positive light and give further support for the publication of this paper.

Kind regards